# Effect of Rotational Velocity on Mechanical and Corrosion Properties of Friction Stir-Welded SUS301L Stainless Steel

**DOI:** 10.3390/ma17143486

**Published:** 2024-07-14

**Authors:** Jianing Dong, Yuming Xie, Shengnan Hu, Junchen Li, Yaobang Zhao, Xiangchen Meng, Yongxian Huang

**Affiliations:** 1State Key Laboratory of Precision Welding & Joining of Materials and Structures, Harbin Institute of Technology, Harbin 150001, China; 2Zhengzhou Research Institute, Harbin Institute of Technology, Zhengzhou 450046, China; 3Shanghai Spaceflight Precision Machinery Institute, Shanghai 201600, China

**Keywords:** stainless steel, friction stir welding, rotational velocity, microstructures, mechanical properties, corrosion resistance

## Abstract

Friction stir welding was utilized to obtain high-quality SUS301L stainless steel joints, whose mechanical and corrosion properties were thoroughly evaluated. Sound joints were obtained with a wide range of rotational velocities from 400 to 700 rpm. The microstructures of the stir zone primarily consisted of austenite and lath martensite without the formation of detrimental phases. The ultimate tensile strength of the welded joints improved with higher rotational velocities apart from 400 rpm. The ultimate tensile strength reached 813 ± 16 MPa, equal to 98.1 ± 1.9% of the base materials (BMs) with a rotational velocity of 700 rpm. The corrosion resistance of the FSW joints was improved, and the corrosion rates related to uniform corrosion with lower rotational velocities were one order of magnitude lower than that of the BMs, which was attributed to the lower martensite content. However, better anti-pitting corrosion performance was obtained with a high rotational velocity of 700 rpm, which was inconsistent with the uniform corrosion results. It could be speculated that a higher martensitic content had a negative effect on the uniform corrosion performance, but had a positive effect on the improvement of the anti-pitting corrosion ability.

## 1. Introduction

Stainless steel, as an important engineering material, is extensively used in various fields, such as rail transportation, petrochemical industries, and medical devices. SUS301L stainless steel is prominently chosen for urban rail transit train production due to its long service life, excellent weldability, high strength, and outstanding corrosion resistance [1]. The base materials (BMs) were obtained in a cold rolled condition. SUS301L stainless steel, a metastable austenitic stainless steel, undergoes martensitic phase transformation induced by plastic deformation during cold rolling, resulting in increased strength and refined grain size. However, fusion welding of stainless steel often results in issues such as the disappearance of deformation martensite, the formation of δ-ferrite within the welded joints, and the precipitation of harmful phases, which could negatively impact the mechanical and corrosion properties of the joints [2,3,4]. Friction stir welding (FSW), a solid-state welding technology, effectively avoids such issues by preventing material melting during the welding process [5,6]. Given the relatively high melting temperature of stainless steel, FSW tools should meet stringent requirements for high-temperature performance [7,8,9]. Designing suitable FSW tools is crucial for the successful execution of the welding process [10,11]. Currently, research into FSW tools for high-melting-point materials is at an immature stage, requiring further investigation [12,13].

In recent years, FSW of stainless steel has become a research hotspot [14,15,16]. Instead of melting, materials experience frictional heat and severe plastic deformation during the FSW process, leading to dynamic recovery and recrystallization, as well as the formation of equiaxed-grain structures [17,18,19]. Sound joints can be obtained through FSW. Micrometer-sized Cr_23_C_6_ precipitates and δ-ferrite were observed in the stir zone (SZ), leading to possibly decreased pitting and intergranular corrosion resistance due to these factors [20]. Park et al. [21,22] conducted FSW on 2 mm and 6 mm thick 304 austenitic stainless steel. In 6 mm-thick plates, the slower cooling rate post-welding resulted in the presence of the σ phase. While 2 mm thick plates showed no presence of the σ phase, δ-ferrite was detected at the austenite grain boundaries in the SZ. Those results indicated a close relationship between the appearance of the σ phase and the formation of δ-ferrite due to the high-temperature transformation of austenite. Wang et al. [23] conducted FSW experiments on 2 mm-thick Fe-18.4Cr-15.8Mn-2.1Mo-0.66N high-nitrogen austenitic stainless steel using polycrystalline cubic boron nitride (pcBN) tools. Sound weld joints were obtained with nitrogen content comparable to the base materials. The SZ primarily consisted of fine austenitic grains and a minor quantity of discontinuously distributed ferrite. Li et al. [14] also found similar austenite-to-ferrite transformations during FSW. Saeid et al. [24] explored the dynamic recrystallization mechanisms in the SZ of 2205 duplex stainless steel joints, concluding that the ferrite phase primarily underwent continuous dynamic recrystallization, while the austenite phase experienced both continuous dynamic and static recrystallization. Emami et al. [25] reported that grain refinement in the thermo-mechanically affected zone (TMAZ) was mainly achieved through recovery in 2205 super duplex stainless steel joints. It was realized by the formation of sub-grain boundaries during extended recovery stages in the SZ. The ferrite phase, with higher stacking fault energy, recrystallized more rapidly, leading to more pronounced grain growth. As of now, numerous scholars have conducted studies on the rotational velocities of the stainless steel welding process. Zang et al. [26] successfully obtained 1060Al-SUS304 steel joints by FSW with a fixed rotational velocity of 800 rpm. Other research [27] has been conducted on rotational velocities of 200 rpm, 600 rpm, and 1000 rpm for 304SS-2219Al joints. Hua et al. [28] investigated the microstructure and properties of 12Cr heat-resistant ferritic steel welded using FSW with different rotational velocities. They determined that the appropriate rotational velocity to produce sound joints should be from 400 rpm up to 800 rpm when the welding speed is 200 mm/min. Furthermore, achieving sound joints in FSW stainless steel also depends on the choice of welding tool materials The welding tool must be made of hard materials with superior thermal resistance and wear resistance at temperatures higher than 800 °C. Although successful welds have been achieved using pcBN and W-25Re, the high costs limited their widespread application.

In this study, we utilized FSW on SUS301L stainless steel based on WC-W composite welding tools, sparked by low-cost welding tools and excellent joint performance. Microstructural evolution, mechanical performances, and corrosion resistance with different rotational velocities were investigated.

## 2. Materials and Methods

Sheets of 2 mm-thick cold rolled SUS301L stainless steel were used. The composition and mechanical properties of SUS301L are listed in Table 1 and Table 2. A split-type FSW tool was selected as shown in Figure 1, with the working part made of prismatic tungsten carbide-reinforced tungsten matrix composites (WC-W composites) and the fixture part made of H13 steel. The shoulder diameter was 16 mm, the pin length was 1.5 mm, and the pin had a prismatic shape. The plunge depth was set to 0.15 mm. The tilt angle was 2°, the welding speed was 70 mm/min, and the rotational velocities were 400 rpm, 500 rpm, 600 rpm, and 700 rpm. A dwelling time of 20 s for preheating was applied before welding.

Metallographic analysis: A specimen approximately 25 mm in length and 5 mm in width was cut perpendicular to the welding direction, as shown in Figure 2a. The specimen was etched for 5 min using a corrosion solution with an HF:HNO_3_ volume ratio of 1:5:44. Cross-sectional morphology and different regions were imaged using a Keyence VHX-1000E optical microscope for image acquisition. Scanning electron microscope (SEM) analysis: SEM with secondary electron imaging was used to observe the surface morphology of the welded joint and the fracture surfaces of tensile test specimens, as well as the surface morphology after the corrosion test. Energy dispersive spectroscopy (EDS) was employed for point and area scanning at positions requiring elemental analysis.

Phase analysis: A specimen with dimensions of 10 × 10 mm^2^ was cut from the center of the weld joints by electric wire cutting. The polished sample underwent ultrasonic cleaning. Subsequently, X-ray diffraction (XRD) analysis was performed to determine the phase composition of the joint. The X-ray source used was Cu Kα, with a scanning range of 2θ = 40 ~ 105°. Data obtained from the analysis were processed by MDI Jade 6.5 software.

Electron backscatter diffraction (EBSD) analysis: In the EBSD analysis, a Hitachi SU5000 SEM equipped with an Oxford probe was utilized. The operating voltage was set at 20 kV with a scanning step of 0.3 μm. The specimen preparation method for EBSD was identical to that used for metallographic specimens. Electrolytic polishing was performed in a solution of 10 vol.% HClO_4_ + 90 vol.% CH_3_COOH at 25 V for 40 s. EBSD data were analyzed using HKL Channel 5 software.

Tensile test: Tensile specimens of the welded joints were prepared as shown in Figure 2a,b. The tensile rate was maintained at 5 mm/min with three specimens tested for each joint, and the final results were averaged.

Electrochemical corrosion test: Specimens were cut from the center of the welded joints. A 3.5% NaCl aqueous solution was prepared as the testing solution, with a saturated calomel electrode as the reference electrode and a platinum sheet as the auxiliary electrode. The exposed area in the solution was 0.25 cm^2^. The specimen served as the working electrode. Polarization curves were measured using a CHI 760E potentiostat. Electrochemical impedance spectroscopy was conducted using a PARSTAT 4000A single-channel potentiostat.

## 3. Results

The surface formation of the joints obtained with different rotational velocities is shown in Figure 3. Welding experiments were conducted with a welding speed of 70 mm/min and rotational velocities of 400 rpm, 500 rpm, 600 rpm, and 700 rpm. FSW produced weld joints with sound surface formation within the range of the experimental parameters. The joints were free of typical FSW defects such as grooves and peels. Additionally, flashes caused by the extrusion effect of the shoulder were not obvious, indicating that the internal formation of the joints was also sound.

Figure 4a shows the surface formation of a 300 mm-length joint produced by the split-type welding tool. The surface quality also appeared to be quite sound, demonstrating that the WC-W composite was a feasible alternative for welding stainless steel. The dimensions with key positions of the welding tool were measured to characterize the wear rate of the tool after discontinuous welding of 11 m-length welding joints, as shown in Figure 4b. Figure 4c shows the dimensional changes at key positions of the welding tool, with all wear rates not exceeding 7%. Overall, the degree of wear was minimal and met the requirements for high-quality weld joint formation during the welding process. The shoulder area of the tungsten-based composite welding tool was characterized by SEM, as shown in Figure 4d,e. No radial or other randomly occurring micro-cracks were observed in various regions of the shoulder. This indicated that the WC-W composite welding tool had relatively outstanding wear resistance during welding, making it suitable for FSW welding of high-melting-point metals such as stainless steel.

Figure 5 depicts the microstructures of different regions of the SZ. The SZ was primarily composed of austenite and lath martensite. According to research by Emami et al. [25], the occurrence of the σ phase led to an evident decrease in corrosion performance in the precipitated regions. As such, desirable corrosion resistance might be obtained in the FSW SUS301 stainless steel joints judging by the microstructures of the SZ.

Figure 6 presents the XRD results for the BMs and the SZ, which mainly consisted of austenite and martensite. The intensity of the austenite (111) peak and the martensite (110) peak changed with varying rotational velocities. These peak intensities show noticeable differences compared to the BMs, indicating a mutual transformation between austenite and martensite during the welding process [29,30]. The proportions of these phases with different rotational velocities were further elucidated in the subsequent EBSD phase diagrams. The σ phase was not found in the XRD results, indicating that the σ phase was inhibited during FSW. However, there is also a possibility that the σ phase exists in concentrations below the detection limit without detection by XRD. In actuality, FSW is featured by a high strain rate and low heat input. Severe plastic deformation induced by a high strain rate brought about the homogenization of the materials, which contributed to the dissolution of the pristine σ phase. In addition, a low heat input during the FSW process can suppress the re-precipitation of these detrimental phases during the post-welding cooling stage. As such, even if there are some fragmented σ phases which exist in the materials but cannot be detected by XRD, they are distributed intragranularly due to the fragmentation induced by severe plastic deformation and the corresponding dynamic recrystallization/recovery, which are not harmful to the corrosion resistance of the FSW joints [31,32].

Figure 7 shows the EBSD orientation maps of the SZs at different rotational velocities. Different grain colors represented various grain orientations. The overall change was not significant, although the grain sizes were with different rotational velocities. The statistics of grain size are illustrated in Figure 7. The grain size in the SZ increased compared to the BMs, primarily due to the welding heat input promoting grain growth [33]. The grain size did not shown a consistent increasing trend, despite a continual rise in heat input as the rotational velocities increased. This phenomenon could be explained by the interaction between heat input and strain rate [16,34]. The heat input rose when the rotational velocity increased, which encouraged grain growth through recrystallization [35]. Simultaneously, the increase in strain rate promoted nucleation during dynamic recrystallization, leading to grain refinement. The final grain size resulted from the competition between these two mechanisms. Grain refinement was mainly attributed to the higher strain rate induced by higher rotational velocities, from 400 rpm to 500 rpm. However, the strain rate effect on grain refinement became secondary as the rotational velocity increased further from 500 rpm to 700 rpm, and the increased heat input led to grain enlargement.

Figure 8a–e depicts EBSD phase maps reflecting the distribution of austenite and martensite fractions with various rotational velocities. The phase fraction in the SZ did not differ significantly below 700 rpm. However, there was a notable increase in martensite content upon reaching 700 rpm. Figure 7f illustrates the statistical changes in phase content with different rotational velocities. The austenite fraction initially rose and then declined with the increase of rotational velocities, while the martensite fraction initially decreased and subsequently increased. As previously analyzed, the changes in phase fraction were influenced by deformation and phase transformation induced by temperature variations. The deformation increased with higher rotational velocities, promoting the transformation of austenite to martensite. Conversely, higher heat input, associated with increased rotational velocities, inhibits this transformation. The heat input predominantly governed the phase fraction changes when the rotational velocity increased from 400 rpm to 500 rpm, resulting in a decrease in the martensite fraction [16]. During the increase in rotational velocity from 500 rpm to 700 rpm, the dominant factor affecting the phase ratio changes became deformation. Consequently, the martensite fraction showed a continuous increase, with the deformation increasing with higher rotational velocities.

Figure 9 shows the kernel average misorientation (KAM) maps of the BMs and the joints with different rotational velocities. The values of KAM were positively related to the density of dislocations. The average KAM (KAM_avg_) values of the joints increased with the increase of rotational velocities, while all the KAM_avg_ values were lower than those of the BMs. This could be attributed to the reduction of martensite content and the growth of grain sizes. Dislocations were depleted during the transformation of martensite into austenite at the post-welding cooling stage. By contrast, the dynamic recovery related to the increased grain sizes in the SZ also consumed dislocations. The KAM_avg_ value of the dislocation was decreased under the combined influence of these two factors. 

Figure 10 presents the tensile properties of the welded joints. As shown in Figure 10b, the ultimate tensile strength initially decreased and then increased with increasing rotational velocity. The grain size decreased from 9.00 μm to 8.71 μm when the rotational velocity increased from 400 rpm to 500 rpm. Although grain refinement tended to increase the ultimate tensile strength, there was a simultaneous decrease in martensite content, which negatively impacted the joint performance. The slight reduction in ultimate tensile strength indicated that the proportion of martensite predominantly influenced the change in ultimate tensile strength. The grain size and the martensite fraction continued to increase as the rotational velocity increased from 500 rpm to 700 rpm, leading to a gradual improvement in joint ultimate tensile strength. The ultimate tensile strength was at its lowest at 500 rpm, representing 95.2 ± 0.8% of the BMs. Conversely, the ultimate tensile strength of the welded joint reached its peak at 813 ± 16 MPa when the rotational velocity was 700 rpm, accounting for 98.1 ± 1.9% of the BMs. Additionally, the ultimate tensile properties of all joints achieved 95% of the BMs, indicating that FSW is a feasible alternative for joining stainless steel. 

Figure 11 shows the fracture morphology of tensile specimens at different magnifications with various rotational velocities. Upon observation, it was evident that the macroscopic fracture surfaces of the joints exhibit a certain degree of necking. The overall fracture was relatively uniform, without any noticeable delamination. Microstructural examination of the fracture surfaces revealed numerous dimples, indicating that the joints possessed a desirable ductility. The tensile fracture mode was therefore characterized as a ductile fracture.

Figure 12a presents the polarization curves of FSW joints and BMs in a 3.5% NaCl aqueous solution with different rotational velocities. Table 3 summarizes the electrochemical response data derived from the polarization curves. Corrosion potentials (*E_corr_*), corrosion current density (*I_corr_)*, pitting potentials (*E_b_*), and protection potentials (*E_rp_*) were calculated. The *I_corr_* values of the FSW joints were one order of magnitude lower than that of the BMs, elucidating that the FSW joints had better corrosion resistance. Moreover, the *E_corr_* values of the FSW joints were nobler with lower rotational velocities. This phenomenon could also be seen in the electrochemical impedance spectroscopy (EIS) results in Figure 12b,c. A larger capacitive arc radius indicated stronger stability of the passive film. Higher equivalent impedance was observed with lower rotational velocities, indicating that these joints have a lower tendency to being corroded. However, one should note that although the *E_corr_* value of the joint with a rotational velocity of 700 rpm was less noble, the value of *E_b_* − *E_rp_* of this joint reached the lowest 0.314 V among all the joints with different rotational velocities. This indicated that the joints with higher rotational velocities and high martensite content were related to a higher pitting corrosion resistance and better ability to repair the passive film on the surface of the joints. Therefore, it could be speculated that higher martensitic content had a negative effect on the uniform corrosion performance, but had a positive effect on the improvement of the anti-pitting corrosion ability. In other words, the rotational velocities need to be compromised to obtain the simultaneous increase in uniform corrosion and pitting corrosion resistance of the FSW joints.

Table 4 presents the fitting results of the electrochemical impedance spectroscopy (EIS) curves, highlighting that the resistance of the passive film significantly surpassed the charge transfer resistance. Passive film resistance played a more crucial role in maintaining stability [36,37,38,39]. The equivalent impedance of the passive film for each joint was calculated based on the data in Table 4. Due to the small charge transfer resistance, primarily reflected in the overall trend were the changes in passive film resistance. The same trend was largely consistent with the changes in the capacitive arc radius and pitting potential. The corrosion resistance deteriorated notably due to grain coarsening with a rotational velocity of 400 rpm. As the rotational velocities increased further, the austenite content increased, and the decrease of the grain size enhanced the corrosion resistance. The highest equivalent impedance was obtained with a rotational velocity of 500 rpm. The austenite content decreased when the rotational velocity reached 700 rpm, leading to a substantial reduction in corrosion resistance.

## 4. Conclusions

Sound FSW joints of SUS301L stainless steel were achieved across various rotational velocities. The microstructure of the SZ mainly consisted of austenite and lath martensite. The formation of the detrimental phases was effectively avoided. The ultimate tensile strength of the welded joints increased with higher rotational velocities. The ultimate tensile strength reached 813 ± 16 MPa, achieving 98.1 ± 1.9% of the BMs with a rotational velocity of 700 rpm. The corrosion resistance of the FSW joints was improved, and the corrosion rates related to uniform corrosion were lower with lower rotational velocities (one order of magnitude lower than the BMs), which was attributed to the lower martensite content. However, better anti-pitting corrosion performance was obtained with a high rotational velocity of 700 rpm, which was inconsistent with the uniform corrosion results. It could be speculated that a higher martensitic content had a negative effect on the uniform corrosion performance, but had a positive effect on the improvement of anti-pitting corrosion ability. Overall, it was necessary to take into account the mechanical and corrosion properties (both uniform corrosion and pitting corrosion) when selecting the welding parameters.

## Figures and Tables

**Figure 1 materials-17-03486-f001:**
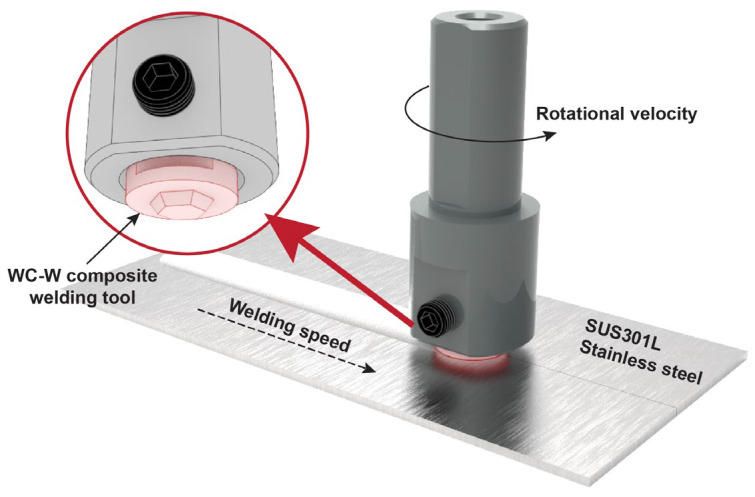
Schematic diagram of friction stir welding of SUS301 stainless steel.

**Figure 2 materials-17-03486-f002:**
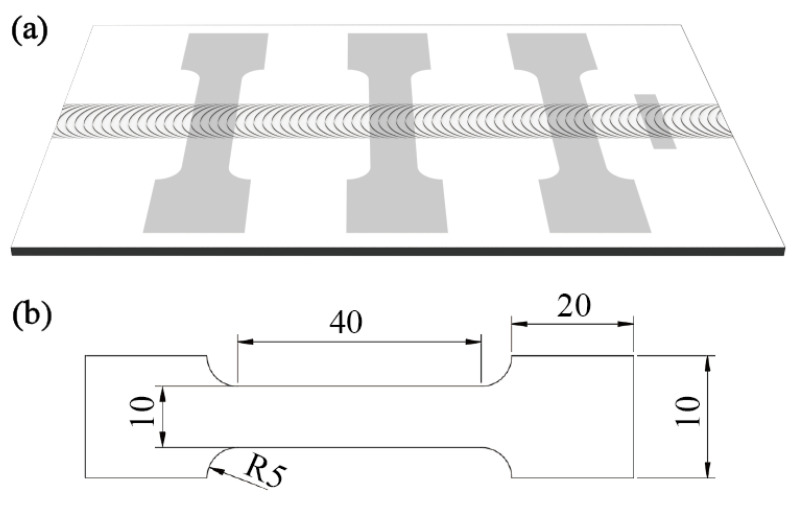
Sampling locations and dimensions: (**a**) specimen locations and (**b**) dimensions of tensile specimens (mm).

**Figure 3 materials-17-03486-f003:**
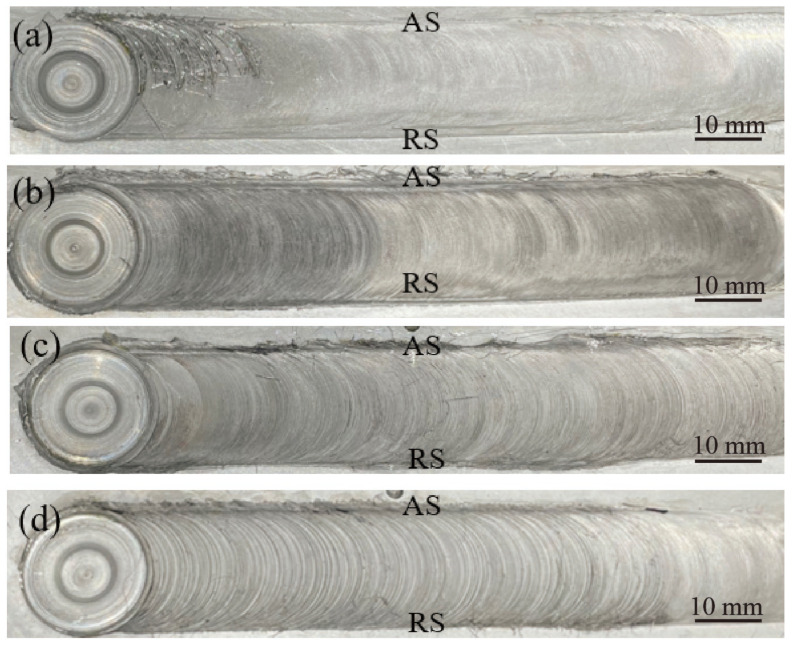
Joint surface formation with different rotational velocities: (**a**) 400 rpm, (**b**) 500 rpm, (**c**) 600 rpm, and (**d**) 700 rpm.

**Figure 4 materials-17-03486-f004:**
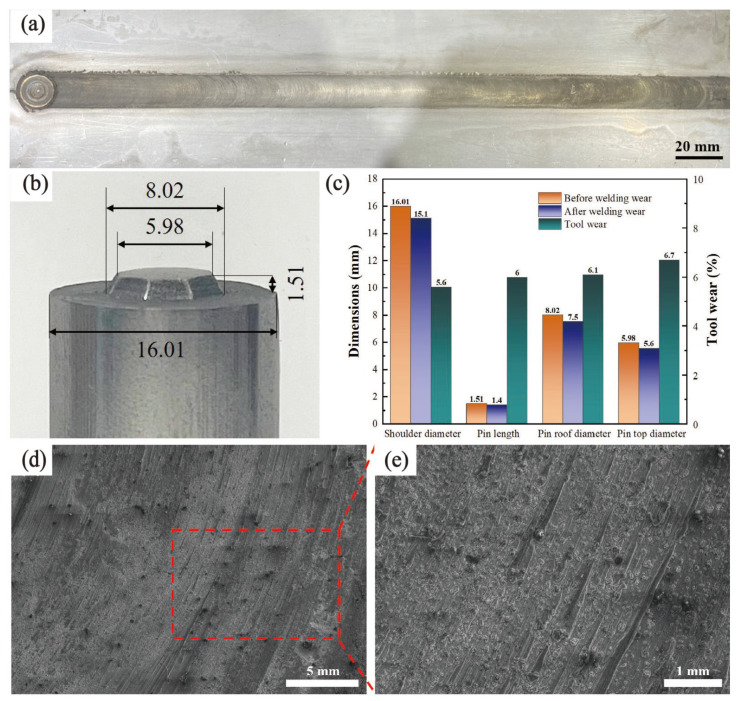
Evaluation of durability of the split-type welding tools: (**a**) partial 300 mm-length joint, (**b**) the diameter before the tool was used (unit: mm), (**c**) diameter changes before and after use, (**d**) SEM image of the tool shoulder area (50× magnification), and (**e**) localized SEM image (200× magnification).

**Figure 5 materials-17-03486-f005:**
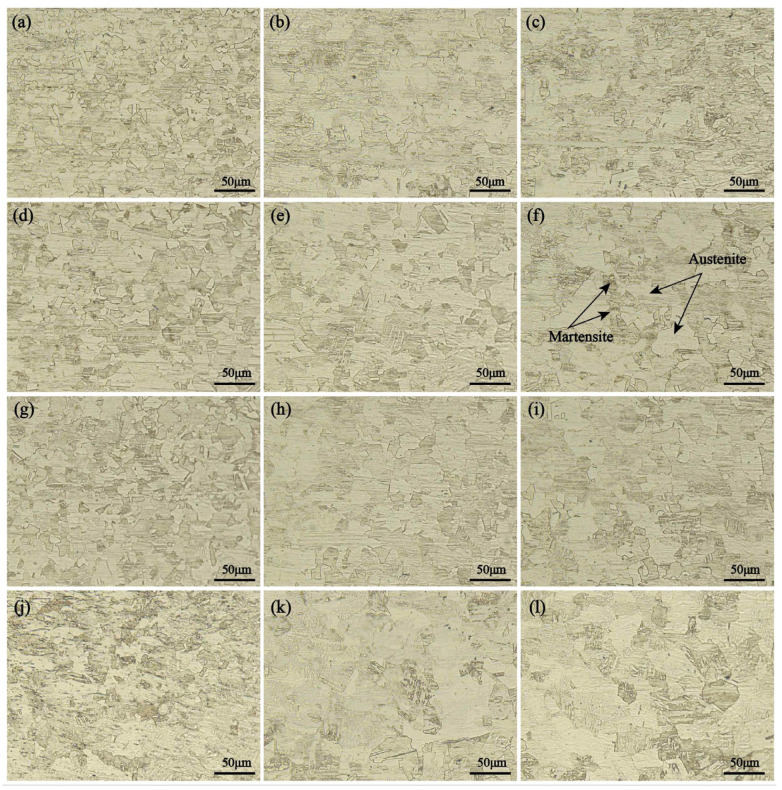
Microstructures of the SZ with different rotational velocities: (**a**) face-400 rpm, (**b**) medium-400 rpm, (**c**) root-400 rpm, (**d**) face-500 rpm, (**e**) medium-500 rpm, (**f**) root-500 rpm, (**g**) face-600 rpm, (**h**) medium-600 rpm, (**i**) root-600 rpm, (**j**) face-700 rpm, (**k**) medium-700 rpm, and (**l**) root-700 rpm.

**Figure 6 materials-17-03486-f006:**
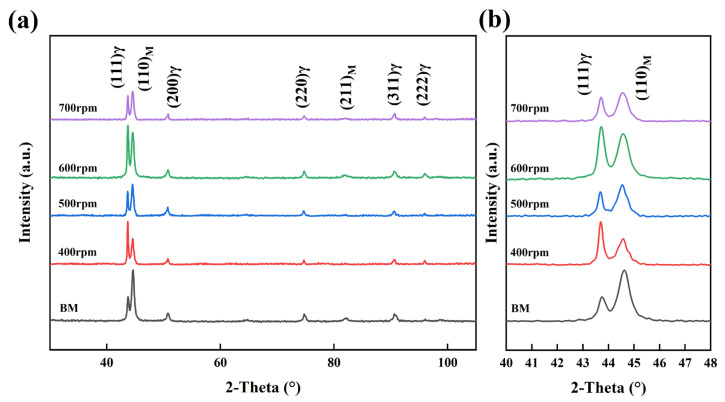
XRD results of BMs and SZs with different rotational velocities (where “M” represents martensite and “γ” represents austenite): (**a**) overall view and (**b**) local magnification.

**Figure 7 materials-17-03486-f007:**
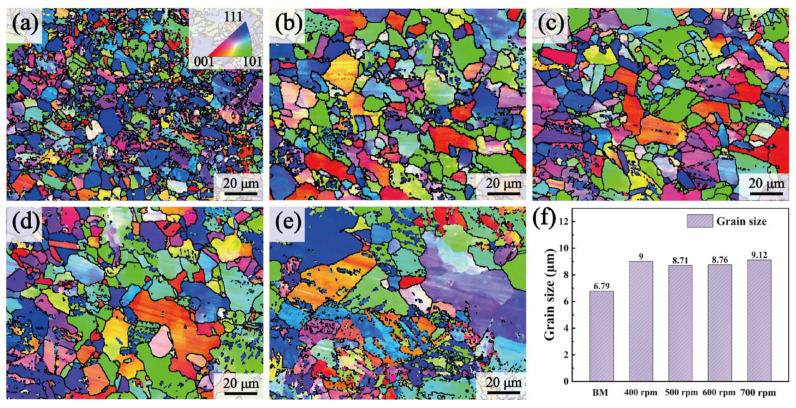
EBSD inversed pole figure with different rotational velocities: (**a**) BMs, (**b**) 400 rpm, (**c**) 500 rpm, (**d**) 600 rpm, (**e**) 700 rpm, and (**f**) average grain sizes.

**Figure 8 materials-17-03486-f008:**
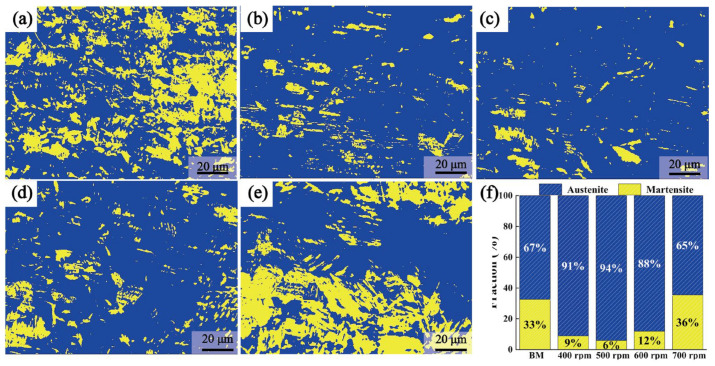
EBSD phase distribution (blue: austenite, yellow: martensite) and two-phase transformation: (**a**) BMs, (**b**) 400 rpm, (**c**) 500 rpm, (**d**) 600 rpm, (**e**) 700 rpm, and (**f**) average grain size of BMs and SZs.

**Figure 9 materials-17-03486-f009:**
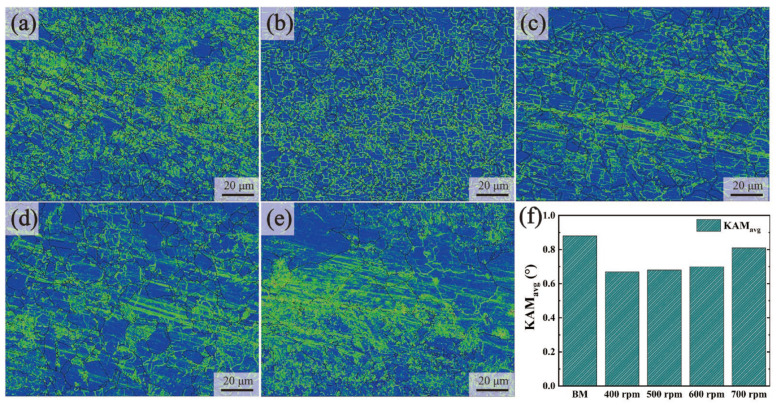
EBSD results of KAM maps (Green means higher misorientation and blue mean lower misorientation): (**a**) BMs, (**b**) 400 rpm, (**c**) 500 rpm, (**d**) 600 rpm, (**e**) 700 rpm, and (**f**) KAM_avg_ of BMs and SZs.

**Figure 10 materials-17-03486-f010:**
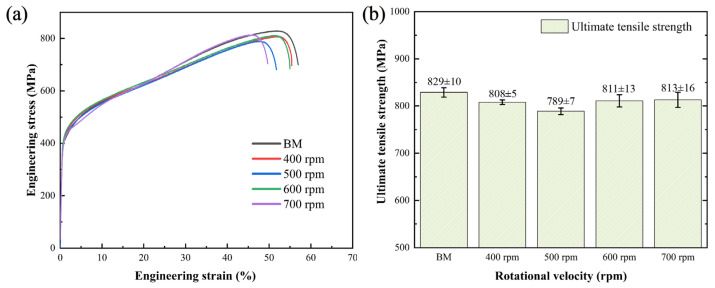
Tensile properties of the joints: (**a**) engineering stress-strain curves and (**b**) ultimate tensile strength.

**Figure 11 materials-17-03486-f011:**
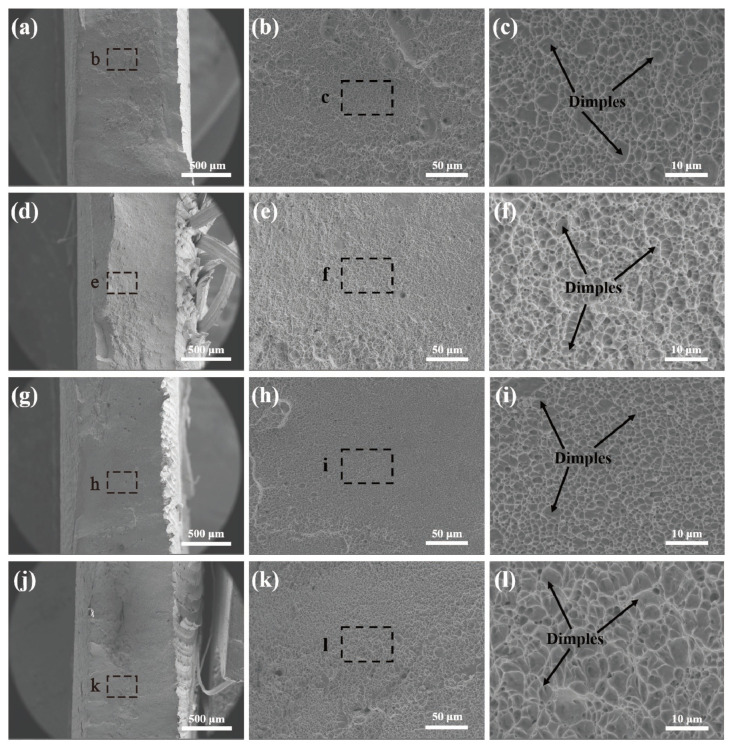
Fractography images of joints: (**a**–**c**) 400 rpm, (**d**–**f**) 500 rpm of (**e**), (**g**–**i**) 600 rpm, and (**j**–**l**) 700 rpm.

**Figure 12 materials-17-03486-f012:**
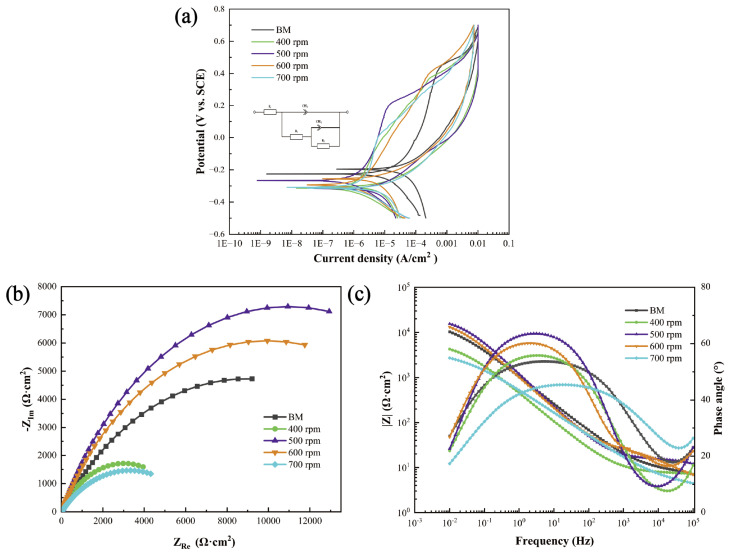
Electrochemical test curves: (**a**) cyclic polarization curves of BMs and joints with different rotational velocities, (**b**) Bode plot, and (**c**) Nyquist plot.

**Table 1 materials-17-03486-t001:** Composition of SUS301L stainless steel (wt.%).

Element	C	Si	Mn	P	S	Ni	Cr	N	Fe
Mass fraction	0.02	0.5	1.0	0.25	0.001	7.0	17.1	0.14	Bal.

**Table 2 materials-17-03486-t002:** Mechanical properties of SUS301L stainless steel.

Material	Ultimate Tensile Strength (MPa)	Yield Strength (MPa)	Elongation (%)	Microhardness (HV)
SUS301L	829	385	57	215

**Table 3 materials-17-03486-t003:** Electrochemical corrosion parameters of BMs and joints with different rotational velocities.

Samples	*E*_corr_ (V vs. SCE)	*I_corr_* (A/cm^2^)	*E_b_* (V vs. SCE)	*E_rp_* (V vs. SCE)	*E_b_ − E_rp_* (V)
BMs	−0.343	4.478 × 10^−5^	0.440	−0.183	0.623
400 rpm	−0.301	1.486 × 10^−6^	0.407	−0.249	0.656
500 rpm	−0.305	1.932 × 10^−6^	0.215	−0.309	0.524
600 rpm	−0.294	1.567 × 10^−6^	0.404	−0.249	0.653
700 rpm	−0.328	2.183 × 10^−6^	0.011	−0.303	0.314

**Table 4 materials-17-03486-t004:** Equivalent circuit fitting data for BMs and joints with different rotational velocities.

Samples	R_s_ (Ω·cm^2^)	CPE_1_ (Ω^−1^·cm^−2^·s^-n^)	n_1_	R_1_ (Ω·cm^2^)	CPE_2_ (Ω^−1^·cm^−2^·s^-n^)	n_2_	R_2_ (Ω·cm^2^)	R_t_ (Ω·cm^2^)
BMs	4.91	3.15 × 10^−5^	0.73	8.67	2.93 × 10^−4^	0.62	17,910	17,918
400 rpm	6.96	8.47 × 10^−5^	0.95	6.20	6.32 × 10^−4^	0.66	6585	6591
500 rpm	10.52	2.70 × 10^−5^	0.75	8.69	5.30 × 10^−4^	0.74	22,800	22,808
600 rpm	7.23	6.49 × 10^−5^	0.62	25.61	2.03 × 10^−4^	0.73	19,970	19,995
700 rpm	8.03	5.85 × 10^−4^	0.98	6.59	3.89 × 10^−4^	0.54	6042	6048

## Data Availability

The data presented in this study are available on request from the corresponding author. The data are not publicly available due to privacy restrictions.

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
