# Peer review of "Effect of Rotational Velocity on Mechanical and Corrosion Properties of Friction Stir-Welded SUS301L Stainless Steel"

_materials, 2024, doi:10.3390/ma17143486_

Round 1
Reviewer 1 Report
Comments and Suggestions for Authors
Materials and methods
1. Please, could you explain how were selected the welding speed (70 mm.min-1) and rotational speeds set (400 to 700 rpm)?
Results
1. Please provide a reference that supports this: “These peak intensities show noticeable differences compared to the BM, indicating a mutual transformation between austenite and martensite during the welding process.”
2. The detection of σ phase is due to the absence of this or low volume fraction that it was not possible to measure those with XRD techniques? It is necessary to discuss this and support it with references.
3. Figure 4: The used magnification is not adequate to observe clearly morphology details of phases.
4. In Figure 6 details of Orientation Map Distribution are not distinguished. Similarly it occurs with the grain size distribution. For this last, it is necessary to add the variance bars of measurements.
5. Page 7 – lines 178 – 181. The sentence needs support references.
6. In Figure 7 please check the used colors to distinguish the phases. These differ in the Distribution maps (blue and yellow) and the bars graphic (blue and orange).
7. Page 7 – lines 192 to 193. Are the grain sizes 9.00 and 8.71 um statistically different?
8. Page 10 – line 249: The figure 12 does not exist. Please, revise it. That figure it is necessary to support the arguments of the subsequence lines, additionally, the paragraph between the lines 249 to 256 needs supporting references.
Comments on the Quality of English LanguageEnglish grammar should be revised in the entire paper.
Author Response
Comment 1: Please, could you explain how were selected the welding speed (70 mm.min-1) and rotational velocitys set (400 to 700 rpm)?
Response to Reviewer 1 comment 1: Thanks for your valuable suggestion, which is instrumental in improving the quality of our paper. We conducted numerous exploratory experiments, with defect-free formation and revealing optimal joint performance at 70 mm/min, achieved between 400-700 rpm. Therefore, we have selected this scope for detailed investigation and discussion.
Comment 2: Please provide a reference that supports this: “These peak intensities show noticeable differences compared to the BM, indicating a mutual transformation between austenite and martensite during the welding process.”
Response to Reviewer 1 comment 2: We have provided a reference to the statement regarding peak intensities indicating mutual transformation between austenite and martensite during the welding process.
- Emami, S.; Saeid, T.; Abdollah-zadeh, A. Effect of Friction Stir Welding Parameters on the Microstructure and Microtexture Evolution of SAF 2205 Stainless Steel. J. Alloys Compd. 2019, 810, 151797, doi:10.1016/j.jallcom.2019.151797.
- Cizek, P.; Whiteman, J.A.; Rainforth, W.M.; Beynon, J.H. EBSD and TEM Investigation of the Hot Deformation Substructure Characteristics of a Type 316L Austenitic Stainless Steel. J. Microsc. 2004, 213, 285–295, doi.org/10.1111/j.0022-2720.2004.01305.x.
Comment 3: The detection of σ phase is due to the absence of this or low volume fraction that it was not possible to measure those with XRD techniques? It is necessary to discuss this and support it with references.
Response to Reviewer 1 comment 3: The σ phase was not found in the XRD results, indicating that the σ phase was inhibited during FSW. However, there is also a possibility that the σ phase exist in concentrations below the detection limit without detection by XRD. In actual, FSW was featured by high strain rate and low heat input. Severe plastic deformation induced by high strain rate brought about the homogenization of the materials, which contributed to the dissolution of the pristine σ phase. Besides, low heat input during the FSW process can suppress the re-precipitation of these detrimental phases during the post-welding cooling stage. As such, even if there are some fragmented σ phases which exist in the materials but cannot be detected by XRD, they are distributed intragranularly due to the fragmentation induced by severe plastic deformation and corresponding dynamic recrystallization/recovery, which are not harmful to the corrosion resistance of the FSWed joints [31,32].
- Wang, J.; Liu, Z.; Tian, H.; Han, P.; Gonzalez-Garcia, Y. Inhibition of the Second Phase Precipitation and Improvement of Intergranular Corrosion Resistance by Boron Segregation at the Grain Boundary of S31254 Superaustenitic Stainless Steel. Corros. Commun. 2024, 15, 1–12, doi:10.1016/j.corcom.2023.08.006.
- Gao, J.; Ma, J.; Yang, S.; Guo, Z.; Ma, J.; Li, H.; Jiang, Z.; Han, P. Grain Boundary Co-Segregation of B and Ce Hindering the Precipitates of S31254 Super Austenitic Stainless Steel. J. Mater. Res. Technol. 2023, 24, 2653–2667, doi:10.1016/j.jmrt.2023.03.135.
Comment 4: Figure 4: The used magnification is not adequate to observe clearly morphology details of phases.
Response to Reviewer 1 comment 4: We have adjusted the magnification of microscope to ensure a clear visualization of phase morphology details and updated the relevant images in the revised version.
Comment 5: In Figure 6 details of Orientation Map Distribution are not distinguished. Similarly it occurs with the grain size distribution. For this last, it is necessary to add the variance bars of measurements.
Response to Reviewer 1 comment 5: We have added the variance bars of measurements in Figure 6f and enhanced the clarity of the orientation map distribution in Figure 7a to improve data presentation.
Comment 6: Page 7 – lines 178 – 181. The sentence needs support references.
Response to Reviewer 1 comment 6: We cited the references in this section to make the statement more convincing. Sabooni et al. [16] reported that the control of the heat input is very important during FSW of UFG stainless steels as it affects some processes such as grain growth, distortion, and phase transformation.
- Sabooni, S.; Karimzadeh, F.; Enayati, M.H.; Ngan, A.H.W. Friction-Stir Welding of Ultrafine Grained Austenitic 304L Stainless Steel Produced by Martensitic Thermomechanical Processing. Mater. Des. 2015, 76, 130–140, doi:10.1016/j.matdes.2015.03.052.
Comment 7: In Figure 7 please check the used colors to distinguish the phases. These differ in the Distribution maps (blue and yellow) and the bars graphic (blue and orange).
Response to Reviewer 1 comment 7: We have adjusted the bar graphic colors to match those in the distribution maps (blue: austenite and yellow: martensite), as shown in Figure 8f. These changes ensured clarity and consistency in visual representation throughout the figure.
Comment 8: Page 7 – lines 192 to 193. Are the grain sizes 9.00 and 8.71 um statistically different?
Response to Reviewer 1 comment 8: We observed minimal differences in grain size under the four experimental parameters as shown in Figure 6f. However, it qualitatively reflected the trend of grain size variation with continuous rotational velocities.
Comment 9: Comments on the Quality of English Language, English grammar should be revised in the entire paper.
Response to Reviewer 1 comment 9: We have carefully revised the entire manuscript to improve the English language and grammar.
Reviewer 2 Report
Comments and Suggestions for Authors
In this work, authors used friction welding to achieve high-quality joints made of stainless steel (type SUS301L). Mechanical and corrosion properties were thoroughly tested. Tight joints were achieved at different range of rotation speeds (400 to 700 rpm). The microstructures in the mixing zone consisted primarily of austenite and lath martensite without the formation of harmful phases. The ultimate tensile strength of the welded joints reached 813±16 MPa (equal to 98.1±1.9% of the base material). The corrosion potential of the joints is increased by 0.049 V compared to the base materials. The difference between the formation and protection potentials is reduced, indicating improved resistance to corrosive media.
The manuscript is very well written. The topic of the paper is interesting, the paper itself has a good structure, the experiment is well-designed, and all measurements are well-executed. The results of the experimental research are clearly presented and supported with numerous tables and figures. Reading the paper is easy and fluid. Overall manuscript is adhering to Journal standards.
The references are valid, and most abbreviations are listed, except for "pcBN" in line 51.
Figure 1 looks great and very nicely illustrates friction stir welding. Is this the authors' own figure or is it taken from another source? If it is taken from another source, please cite the original source.
My main suggestion to the authors is to revise the abstract as well as the conclusion. The abstract should provide a bit more information to make it more appealing and to attract the scientific community, making the manuscript more visible. Additionally, the conclusion should be better written, highlighting the results obtained in this experimental work more prominently.
Author Response
Comment: In this work, authors used friction welding to achieve high-quality joints made of stainless steel (type SUS301L). Mechanical and corrosion properties were thoroughly tested. Tight joints were achieved at different range of rotational velocities (400 to 700 rpm). The microstructures in the mixing zone consisted primarily of austenite and lath martensite without the formation of harmful phases. The ultimate tensile strength of the welded joints reached 813±16 MPa (equal to 98.1±1.9% of the base material). The corrosion potential of the joints is increased by 0.049 V compared to the base materials. The difference between the formation and protection potentials is reduced, indicating improved resistance to corrosive media.
The manuscript is very well written. The topic of the paper is interesting, the paper itself has a good structure, the experiment is well-designed, and all measurements are well-executed. The results of the experimental research are clearly presented and supported with numerous tables and figures. Reading the paper is easy and fluid. Overall manuscript is adhering to Journal standards.
Response to Reviewer 2 comment: Thanks for your positive comments. It is your valuable comment that helped us improve the manuscript to be published in the prestigious journal Materials.
Comment 1: The references are valid, and most abbreviations are listed, except for "pcBN" in line 51.
Response to Reviewer 2 comment 1: We have added the abbreviations of polycrystalline cubic boron nitride (pcBN) in the Introduction.
Comment 2: Figure 1 looks great and very nicely illustrates friction stir welding. Is this the authors' own figure or is it taken from another source? If it is taken from another source, please cite the source.
Response to Reviewer 2 comment 2: Figure 1 is a schematic diagram of the process that we drew ourselves to illustrate friction stir welding.
Comment 3: My main suggestion to the authors is to revise the abstract as well as the conclusion. The abstract should provide a bit more information to make it more appealing and to attract the scientific community, making the manuscript more visible. Additionally, the conclusion should be better written, highlighting the results obtained in this experimental work more prominently.
Response to Reviewer 2 comment 3: Moreover, we enhanced the conclusion and abstract to improve clarity and coherence. The microstructures of the stir zone primarily consisted of austenite and lath martensite without the formation of detrimental phases. The ultimate tensile strength of the welded joints improved with higher rotational velocities apart from 400 rpm. The ultimate tensile strength reached 813±16 MPa, equal to 98.1±1.9% of the base materials (BM) with a rotational velocity of 700 rpm. The corrosion resistance of the FSWed joints was improved, and the corrosion rate related to uniform corrosion with lower rotational velocities were one order of magnitude lower than that of BM, which was attributed to the lower martensite content. However, better anti-pitting corrosion performance was obtained with high rotational velocity of 700 rpm, which was inconsistent to the results of uniform corrosion results. It could be speculated that higher martensitic contents had a negative effect on the uniform corrosion performances, but had a positive effect on the improvement of anti-pitting corrosion ability.
Reviewer 3 Report
Comments and Suggestions for Authors
All the criticism is presented in the Review Report.

Comments on the Quality of English LanguageFew typos could be fixed.
Author Response
Comment 1: 1. Abstract, Title, and Objectives
The title does not match the appointments presented in the abstract. How about the effect of rotational velocity on the mechanical and corrosion properties? The authors must show the findings related to the rotational velocity, otherwise, the Title must be written again.
Response to Reviewer 3 comment 1: We appreciate your insightful comment regarding the main scientific contribution of the manuscript. We have revised the abstract to emphasize the relationship between rotational velocity, corrosion resistance, and mechanical strength. Sound joints were obtained with a wide range of rotational velocities from 400 to 700 rpm. The microstructures of the stir zone primarily consisted of austenite and lath martensite without the formation of detrimental phases. The ultimate tensile strength of the welded joints improved with higher rotational velocities apart from 400 rpm. The ultimate tensile strength reached 813±16 MPa, equal to 98.1±1.9% of the base materials (BM) with a rotational velocity of 700 rpm. The corrosion resistance of the FSWed joints was improved, and the corrosion rate related to uniform corrosion with lower rotational velocities were one order of magnitude lower than that of BM, which was attributed to the lower martensite content. However, better anti-pitting corrosion performance was obtained with high rotational velocity of 700 rpm, which was inconsistent to the results of uniform corrosion results. It could be speculated that higher martensitic contents had a negative effect on the uniform corrosion performances, but had a positive effect on the improvement of anti-pitting corrosion ability.
Comment 2: let's see the work purpose, at the end of the introduction (lines 67 to 70): “In this study, we utilized FSW on SUS301L stainless steel based on WC-Wcomposite welding tools, sparked by low-cost welding tools and excellent joint performance. Microstructural evolution, mechanical performances, and corrosion resistance were discussed in detail." Once again, no mention is made of the rotational velocity!
Response to Reviewer 3 comment 2: We have mentioned the effect of rotational velocities on microstructural evolution, mechanical performances, and corrosion resistance in the purpose of the study of the manuscript: “Microstructural evolution, mechanical performances, and corrosion resistance with different tool rotational velocities are investigated.” Furthermore, we have supplemented the description and study of rotational velocity in the introduction.
Comment 3: There are a few typos, as examples marked up in the PDF file.
Response to Reviewer 3 comment 3: Thank you for your suggestion, but unfortunately, we did not find the error markings you mentioned in the PDF.
Comment 4: No mention is made of the effect of rotational velocity on the mechanical and corrosion properties and this is an issue investigated in deep. The authors need to review the background of the rotational velocity to support the discussion of the results found.
Response to Reviewer 3 comment 4: We have made appropriate modifications to the introduction section. Zang et al. [26] successfully obtained 1060Al-SUS304 steel joints by FSW at a fixed rotational velocity of 800 rpm. Another research [27] has been conducted on the rotational velocities of 200 rpm, 600 rpm, and 1000 rpm for the 304SS-2219Al joint. Hua et al. [28] investigated the microstructure and properties of the 12Cr heat-resistant ferritic steel welded using FSW with different rotational velocities. They found that, at constant welding speed (200 mm/min), the optimal rotational velocities to produce sound welds should be from 400 rpm up to 800 rpm.
- Zhang, M.; Wang, Y.D.; Xue, P.; Zhang, H.; Ni, D.R.; Wang, K.S.; Ma, Z.Y. High-Quality Dissimilar Friction Stir Welding of Al to Steel with No Contacting between Tool and Steel Plate. Mater. Charact. 2022, 191, doi:10.1016/j.matchar.2022.112128.
- Zhang, M.; Liu, J.M.; Xue, P.; Liu, F.C.; Wu, L.H.; Ni, D.R.; Xiao, B.L.; Wang, K.S.; Ma, Z.Y. Eliminating Cu-Rich Intermetallic Compound Layer in Dissimilar Friction Stir Welding of 304 Stainless Steel and 2219 Al Alloy via Ultralow Rotation Speed. J. Mater. Process. Technol. 2024, 329, doi:10.1016/j.jmatprotec.2024.118444.
- Hua, P.; Moronov, S.; Nie, C.Z.; Sato, Y.S.; Kokawa, H.; Park, S.H.C.; Hirano, S. Microstructure and Properties in Friction Stir Weld of 12Cr Steel. Sci. Technol. Weld. Join. 2014, 19, 76–81, doi:10.1179/1362171813Y.0000000167.
Comment 5: Rotational velocity levels adopted must be presented, justified, and detailed in the materials and methods section, not only the range of velocities (line 78).
Response to Reviewer 3 comment 5: We have supplemented the materials and methods section with specific parameters used in this study: rotational velocities were 400 rpm, 500 rpm, 600 rpm, and 700 rpm.
Comment 6: Lines 82 and 83:There is no schematic drawing to show where the specimens were taken. It is necessary to prepare a scheme of the metallographic specimens' positioning. Besides, no information is given about the specimens of tensile tests (dimensions) and positioning.
Response to Reviewer 3 comment 6: We have included schematic diagrams of the positioning of metallographic specimens for the samples (Figure 2a), as well as schematic diagrams of the positioning of tensile test specimens (Figure 2a), and a schematic diagram depicting the dimensions of the tensile test specimens(Figure 2b), shown in Figure 2.
Comment 7: 3.3 No detail is provided about the specimen preparation for EBSD evaluation (Lines 91,and92).
Response to Reviewer 3 comment 7: In EBSD analysis, a Hitachi SU5000 SEM equipped with an Oxford probe was utilized. The operating voltage was set at 20 kV with a scanning step of 0.3 μm. The specimen preparation method for EBSD was identical to that used for metallographic specimens. A segment approximately 25 mm in length and 5 mm in width was cut perpendicular to the welding direction using electric wire cutting. This specimen was embedded in acrylic resin to create a metallographic specimen. The was polished to remove wire-cutting marks using 400-grit sandpaper, followed by stepwise grinding with 800, 1500, 2000, and 2500-grit sandpaper. Finally, the specimen was polished with 1.5 μm diamond polishing paste until the surface was scratch-free. The specimen was etched for 5 minutes using a corrosion solution with an HF: HNO3 with a volume ratio of 1:5:44. Electrolytic polishing was performed in a solution of 10 vol.% HClO4 + 90 vol.% CH3COOH at 25 V for 40 s. EBSD data were analyzed using HKL Channel 5 software. We have incorporated the specific details into the manuscript.
Comment 8: There is no detail about the electrochemical evaluation. No standard is presented, no electrochemical cell and configuration, and no scheme showing where the specimens were sliced out. It is mandatory to insert and describe the methods used.
Response to Reviewer 3 comment 8: We have incorporated the relevant details of the electrochemical tests into the manuscript. Electric wire cutting was used to cut samples from the center of the welded joints. Non-test areas were sealed with polymethyl methacrylate. The exposed area in the solution was 0.25 cm². A 3.5% NaCl aqueous solution was prepared as the testing solution, with a saturated calomel electrode as the reference electrode and a platinum sheet as the auxiliary electrode. The specimen served as the working electrode. Polarization curves were measured using a CHI 760E potentiostat. Electrochemical impedance spectroscopy was conducted using a PARSTAT 4000A single-channel potentiostat.
Comment 9: Hardness testing is proposed in section 2, but no result of hardness is presented in the manuscript.
Response to Reviewer 3 comment 9: We are sorry for that, this was a mistake in our manuscript writing process. We have removed the hardness testing from the materials and methods section.
Comment 10: From lines 109 to 126, the authors present the welding tool and too| surface images and argue that the tool was not significantly worn, indicating that the material adopted is good for FSW purposes. I don't think this is possible to say and the purpose of the work is not that. If the authors want to discuss this point, they must bring the background about it and adjust the work purpose.
Response to Reviewer 3 comment 10: This paragraph aims to demonstrate that the WC-W composites meet the basic requirements for welding high-melting-point metals such as stainless steel while being low-cost. The focus is not on claiming optimal wear resistance in welding. The manuscript has been appropriately revised in this section to ensure accuracy in its presentation
Comment 11: 4.2 Lines 127-128: “Figure 4 depicts the microstructures of different regions of the SZ. SZ was primarily composed of austenite and lath martensite, without visible ferrite phase and a phase". I think the images are in low magnification, hindering a confidence microstructure description. Also, considering the magnification, it is too risky to state: "without visible ferrite phase and a phase".
Response to Reviewer 3 comment 11: We have enlarged Figure 5 for a clearer and more intuitive observation. Additionally, we fully agree with your point that “stating without visible ferrite phase and a phase is too risky”. Therefore, we have removed this conclusion from the manuscript to ensure greater rigor.
Comment 12:From Figure 6 and Figure 7(EBSD results), becomes clear that a better description of the base metal (BM) should be done at the beginning of the manuscript. The readers need background about SUS301L(Austenitic steel), especially for the production history. It seems that the stee| strip is cold rolled, with 33% martensite in the microstructure (Figure 7). Is it really expected to have such a high amount of martensite in austenitic steels? How about the strain hardening and dislocation substructures? It needs further discussion and detailment. Also, the authors must describe the initial microstructure (BM),bring research works and references for that, and understand how important are these details to explain what is happening with the grain size and martensite fraction after welding, as different rotational velocities are chosen. Note, that the normal rule is to have grain size refinement from FSW welding in the stir zone, exactly the opposite of this work. This is the reason why this is very important to the clarity of the work.
Response to Reviewer 3 comment 12: (1) SUS301L stainless steel, as a metastable austenitic stainless steel, undergoes martensitic phase transformation induced by plastic deformation during cold rolling, which increases its strength and refines the grain size. Typically, grain growth and phase structure transformation occur after secondary processing. (2) The BM was obtained in the cold rolled condition. Due to cold rolling, BM has refined grains and contains a large amount of deformation-induced martensite. (3) Figures 9 show the kernel average misorientation (KAM) maps of the BM and the joint with different rotational velocities. The values of KAM were positively related to the density of dislocations. The average KAM (KAMavg) values of the joints increased with the increase of rotational velocities, while all the KAMavg values were lower than that of BM. This could be attributed to the reduction of martensite content and the growth of grain sizes. Dislocations were depleted during the transformation of martensite into austenite at the post-welding cooling stage. By contrast, the dynamic recovery related to the increased grain sizes in SZ also consumed dislocations. The KAMavg value of the dislocation was decreased under the combined influence of these two factors. (4)Overall, above 400 rpm, the phase ratio changed and heat input rose, resulting in both grain size and martensite content increase with higher rotational velocities. (5)Typically, during the FSW process, severe plastic deformation and subsequent dynamic recrystallization induced by mechanical stirring result in finer grain sizes. However, if the BM possesses ultrafine grain sizes or if excessive heat input occurs during welding, the grain size in the SZ may be larger than the original grain size of the base material.
Comment 13: Grain Size(Figure 6f): Could the authors insert the number of analyzed grains in each evaluated area? How many images were analyzed to have the average value? It seems that the grain size is similar for all welding conditions, roughly about 9 micrometers. I think discussing the effect of welding rotational velocity on the grain size is a bit hard in the present case. The authors could try to differentiate the welding conditions based on the grain size heterogeneities.
Response to Reviewer 3 comment 13:(1) The number of analyzed grains in each evaluated area is as follows: BM:1273, 400 rpm:3411, 500 rpm:4021 600 rpm:6051, and 700 rpm:8561. (2) After analyzing three images for each parameter, we calculated the average. (3) We observed minimal differences in grain size under the four experimental parameters as shown in Figure 6f. However, it qualitatively reflects the trend of grain size variation with continuous rotational velocity.
Comment 14: How many tensile specimens were tested? Here, once again, is important to bring the details pointed out in item 3.2 of this report.
Response to Reviewer 3 comment 14: Each parameter was tested on three tensile specimens, and the average values were taken. We have added detailed information about this in the manuscript.
Comment 14: I could not find the discussion section. The results still need better analysis and the discussion needs to bring the background, citing previous works and debating what is happening in the specific case here with the microstructure, mechanical properties, and corrosion behavior. This is missing, to a certain point, because the introduction did not focus on the effect of rotational velocities on the microstructure, properties, and corrosion behavior. It should be re-written and previous works must be there. Thus, to discuss the results will be a bit easier.
Response to Reviewer 3 comment 14:. We have supplemented the discussion in the manuscript regarding rotational velocity, providing examples from Zang et al.[26, 27]and Hua et al.[28], covering arguments on its influence on microstructure, mechanical properties, and corrosion resistance. We have rewritten the conclusion section of the paper.
- Zhang, M.; Wang, Y.D.; Xue, P.; Zhang, H.; Ni, D.R.; Wang, K.S.; Ma, Z.Y. High-Quality Dissimilar Friction Stir Welding of Al to Steel with No Contacting between Tool and Steel Plate. Mater. Charact. 2022, 191, doi:10.1016/j.matchar.2022.112128.
- Zhang, M.; Liu, J.M.; Xue, P.; Liu, F.C.; Wu, L.H.; Ni, D.R.; Xiao, B.L.; Wang, K.S.; Ma, Z.Y. Eliminating Cu-Rich Intermetallic Compound Layer in Dissimilar Friction Stir Welding of 304 Stainless Steel and 2219 Al Alloy via Ultralow Rotation Speed. J. Mater. Process. Technol. 2024, 329, doi:10.1016/j.jmatprotec.2024.118444.
- Hua, P.; Moronov, S.; Nie, C.Z.; Sato, Y.S.; Kokawa, H.; Park, S.H.C.; Hirano, S. Microstructure and Properties in Friction Stir Weld of 12Cr Steel. Sci. Technol. Weld. Join. 2014, 19, 76–81, doi:10.1179/1362171813Y.0000000167.
Comment 15: This section is not responding to the work purpose, since:
The microstructure needs to be analyzed in depth. From the images (Figure 4) is not possible to conclude about austenite-to-martensite and deleterious phase presence. Also, the authors need to bring the previous description of BM and all production details.
The grain size should be discussed more because the trend of grain refinement in the stir zone of FSW was not observed in the present work. Why?There is a missing corrosion test description, as well as, the possible reasons for the behavior, also dependent on a better microstructure description.
Response to Reviewer 3 comment 15: (1) We adjusted the magnification of Figure 4 (as shown in Figure 5) to allow for a more intuitive observation of the microstructural distribution. (2) We have added the descriptions of the BM in the Introduction section. In the materials and methods, we have added specific rotational velocities parameters and the parameters of all tests. (3) The BM possesses ultrafine grain sizes or if excessive heat input occurs during welding, the grain size in the stirred zone (SZ) may be larger than the original grain size of the BM. (4) We have provided the details of the corrosion tests in the second section and included a discussion on rotational velocity and corrosion resistance in the results section.
Reviewer 4 Report
Comments and Suggestions for Authors
The entire research problem is interesting and is very well presented in the article. Despite the very good quality of the work, I suggest making a few corrections, which in mu opinion much more improve the quality of this work:
#1 Figure 4 - I suggest change the nomenclature. In the caption are used: top, medium and bottom. To be consistent with the nomenclature used in welding, please change: top -> face, bottom -> root
#2 Figure 9 - the pictures are very dark and therefore difficult to read. Please, improve the quality of pictures.
#3 Figure 9 caption - Names of points (a) - (l) must be revised and described by different way. Eg. "(a) - (c) 400 rpm; (d) - (f) 500 rpm .... and add at figures a,d,g,j locations of figures b,e,h,k. It will be more understandable
#4 line 249 - there is no Figure 12
#5 Conclusions are to compressed. There should be some 2-3 sentences of overall sumarization and pointed conclusions.
Author Response
Comment: The entire research problem is interesting and is very well presented in the article. Despite the very good quality of the work, I suggest making a few corrections, which in mu opinion much more improve the quality of this work:
Response to Reviewer 4 comment: Thanks for your positive comments. It is your valuable comment that helped us improve the manuscript to be published in the prestigious journal Materials. In addition, the issues pointed out in the manuscript has been refined and modified.
Comment 1: Figure 4 - I suggest change the nomenclature. In the caption are used: top, medium and bottom. To be consistent with the nomenclature used in welding, please change: top -> face, bottom -> root
Response to Reviewer 4 comment 1: We have renamed Figure 5 to (a) face-400 rpm, (b) medium-400 rpm, (c) root-400 rpm, (d) face-500 rpm, (e) medium-500 rpm, (f) root-500 rpm, (g) face-600 rpm, (h) medium-600 rpm, (i) root-600 rpm, (j) face-700 rpm, (k) medium-700 rpm, and (l) root-700 rpm. to ensure more accurate representation in the manuscript.
Comment 2: Figure 9 - the pictures are very dark and therefore difficult to read. Please, improve the quality of pictures.
Response to Reviewer 4 comment 2: We have adjusted the brightness and contrast of Figure 9 to improve readability.
Comment 3: Figure 9 caption - Names of points (a) - (l) must be revised and described by different way. Eg. "(a) - (c) 400 rpm; (d) - (f) 500 rpm .... and add at figures a,d,g,j locations of figures b,e,h,k. It will be more understandable
Response to Reviewer 4 comment 3: We have corrected it to Figure 11. Fractography images of joints: (a)-(c) 400 rpm, (d)-(f) 500 rpm of e, (g)-(i) 600 rpm, and (j)-(l) 700 rpm.
Comment 4: line 249 - there is no Figure 12
Response to Reviewer 4 comment 4: We have checked and removed this error from the manuscript.
Comment 5: Conclusions are to compressed. There should be some 2-3 sentences of overall sumarization and pointed conclusions.
Response to Reviewer 4 comment 5: We enhanced the conclusion and abstract to improve clarity and coherence. The ultimate tensile strength of the welded joints improved with higher rotational velocities apart from 400 rpm. The ultimate tensile strength reached 813±16 MPa, equal to 98.1±1.9% of the base materials (BM) with a rotational velocity of 700 rpm. The corrosion resistance of the FSWed joints was improved, and the corrosion rate related to uniform corrosion with lower rotational velocities were one order of magnitude lower than that of BM, which was attributed to the lower martensite content. However, better anti-pitting corrosion performance was obtained with high rotational velocity of 700 rpm, which was inconsistent to the results of uniform corrosion results. It could be speculated that higher martensitic contents had a negative effect on the uniform corrosion performances, but had a positive effect on the improvement of anti-pitting corrosion ability.
Round 2
Reviewer 1 Report
Comments and Suggestions for Authors
No comments.
Author Response
We are very sorry that there were some problems in the system of the last version, which caused you to not see our modifications in the manuscript. We have made another change in the system. Please review it again.
Comment 1: Please, could you explain how were selected the welding speed (70 mm.min-1) and rotational velocitys set (400 to 700 rpm)?
Response to Reviewer 1 comment 1: Thanks for your valuable suggestion, which is instrumental in improving the quality of our paper. We conducted numerous exploratory experiments, with defect-free formation and revealing optimal joint performance at 70 mm/min, achieved between 400-700 rpm. Therefore, we have selected this scope for detailed investigation and discussion.
Comment 2: Please provide a reference that supports this: “These peak intensities show noticeable differences compared to the BM, indicating a mutual transformation between austenite and martensite during the welding process.”
Response to Reviewer 1 comment 2: We have provided a reference to the statement regarding peak intensities indicating mutual transformation between austenite and martensite during the welding process.
- Emami, S.; Saeid, T.; Abdollah-zadeh, A. Effect of Friction Stir Welding Parameters on the Microstructure and Microtexture Evolution of SAF 2205 Stainless Steel. J. Alloys Compd. 2019, 810, 151797, doi:10.1016/j.jallcom.2019.151797.
- Cizek, P.; Whiteman, J.A.; Rainforth, W.M.; Beynon, J.H. EBSD and TEM Investigation of the Hot Deformation Substructure Characteristics of a Type 316L Austenitic Stainless Steel. J. Microsc. 2004, 213, 285–295, doi.org/10.1111/j.0022-2720.2004.01305.x.
Comment 3: The detection of σ phase is due to the absence of this or low volume fraction that it was not possible to measure those with XRD techniques? It is necessary to discuss this and support it with references.
Response to Reviewer 1 comment 3: The σ phase was not found in the XRD results, indicating that the σ phase was inhibited during FSW. However, there is also a possibility that the σ phase exist in concentrations below the detection limit without detection by XRD. In actual, FSW was featured by high strain rate and low heat input. Severe plastic deformation induced by high strain rate brought about the homogenization of the materials, which contributed to the dissolution of the pristine σ phase. Besides, low heat input during the FSW process can suppress the re-precipitation of these detrimental phases during the post-welding cooling stage. As such, even if there are some fragmented σ phases which exist in the materials but cannot be detected by XRD, they are distributed intragranularly due to the fragmentation induced by severe plastic deformation and corresponding dynamic recrystallization/recovery, which are not harmful to the corrosion resistance of the FSWed joints [31,32].
- Wang, J.; Liu, Z.; Tian, H.; Han, P.; Gonzalez-Garcia, Y. Inhibition of the Second Phase Precipitation and Improvement of Intergranular Corrosion Resistance by Boron Segregation at the Grain Boundary of S31254 Superaustenitic Stainless Steel. Corros. Commun. 2024, 15, 1–12, doi:10.1016/j.corcom.2023.08.006.
- Gao, J.; Ma, J.; Yang, S.; Guo, Z.; Ma, J.; Li, H.; Jiang, Z.; Han, P. Grain Boundary Co-Segregation of B and Ce Hindering the Precipitates of S31254 Super Austenitic Stainless Steel. J. Mater. Res. Technol. 2023, 24, 2653–2667, doi:10.1016/j.jmrt.2023.03.135.
Comment 4: Figure 4: The used magnification is not adequate to observe clearly morphology details of phases.
Response to Reviewer 1 comment 4: We have adjusted the magnification of microscope (Figure 5) to ensure a clear visualization of phase morphology details and updated the relevant images in the revised version.
Comment 5: In Figure 6 details of Orientation Map Distribution are not distinguished. Similarly it occurs with the grain size distribution. For this last, it is necessary to add the variance bars of measurements.
Response to Reviewer 1 comment 5: We have added the variance bars of measurements in Figure 6f and enhanced the clarity of the orientation map distribution in Figure 7a to improve data presentation.
Comment 6: Page 7 – lines 178 – 181. The sentence needs support references.
Response to Reviewer 1 comment 6: We cited the references in this section to make the statement more convincing. Sabooni et al. [16] reported that the control of the heat input is very important during FSW of UFG stainless steels as it affects some processes such as grain growth, distortion, and phase transformation.
- Sabooni, S.; Karimzadeh, F.; Enayati, M.H.; Ngan, A.H.W. Friction-Stir Welding of Ultrafine Grained Austenitic 304L Stainless Steel Produced by Martensitic Thermomechanical Processing. Mater. Des. 2015, 76, 130–140, doi:10.1016/j.matdes.2015.03.052.
Comment 7: In Figure 7 please check the used colors to distinguish the phases. These differ in the Distribution maps (blue and yellow) and the bars graphic (blue and orange).
Response to Reviewer 1 comment 7: We have adjusted the bar graphic colors to match those in the distribution maps (blue: austenite and yellow: martensite), as shown in Figure 8f. These changes ensured clarity and consistency in visual representation throughout the figure.
Comment 8: Page 7 – lines 192 to 193. Are the grain sizes 9.00 and 8.71 um statistically different?
Response to Reviewer 1 comment 8: We observed minimal differences in grain size under the four experimental parameters as shown in Figure 6f. However, it qualitatively reflected the trend of grain size variation with continuous rotational velocities.
Comment 9: Comments on the Quality of English Language, English grammar should be revised in the entire paper.
Response to Reviewer 1 comment 9: We have carefully revised the entire manuscript to improve the English language and grammar. As suggested by the reviewer, we have corrected some sentences as shown below.
It could be speculated that higher martensitic contents had a negative effect on the uniform corrosion performances but had a positive effect on the improvement of anti-pitting corrosion ability.
Those results indicated a close relationship between the appearance of the σ phase and the formation of δ-ferrite due to the high-temperature transformation of austenite.
The corrosion resistance deteriorated notably due to grain coarsening with a rotational velocity of 400 rpm.

Reviewer 3 Report
Comments and Suggestions for Authors
Comments on the Quality of English LanguageAuthor Response
We are very sorry that there were some problems in the system of the last version, which caused you to not see our modifications in the manuscript. We have made another change in the system. Please review it again.
Comment 1: 1. Abstract, Title, and Objectives
The title does not match the appointments presented in the abstract. How about the effect of rotational velocity on the mechanical and corrosion properties? The authors must show the findings related to the rotational velocity, otherwise, the Title must be written again.
Response to Reviewer 3 comment 1: We appreciate your insightful comment regarding the main scientific contribution of the manuscript. We have revised the abstract to emphasize the relationship between rotational velocity, corrosion resistance, and mechanical strength. Sound joints were obtained with a wide range of rotational velocities from 400 to 700 rpm. The microstructures of the stir zone primarily consisted of austenite and lath martensite without the formation of detrimental phases. The ultimate tensile strength of the welded joints improved with higher rotational velocities apart from 400 rpm. The ultimate tensile strength reached 813±16 MPa, equal to 98.1±1.9% of the base materials (BM) with a rotational velocity of 700 rpm. The corrosion resistance of the FSWed joints was improved, and the corrosion rate related to uniform corrosion with lower rotational velocities were one order of magnitude lower than that of BM, which was attributed to the lower martensite content. However, better anti-pitting corrosion performance was obtained with high rotational velocity of 700 rpm, which was inconsistent to the results of uniform corrosion results. It could be speculated that higher martensitic contents had a negative effect on the uniform corrosion performances, but had a positive effect on the improvement of anti-pitting corrosion ability.
Comment 2: let's see the work purpose, at the end of the introduction (lines 67 to 70): “In this study, we utilized FSW on SUS301L stainless steel based on WC-Wcomposite welding tools, sparked by low-cost welding tools and excellent joint performance. Microstructural evolution, mechanical performances, and corrosion resistance were discussed in detail." Once again, no mention is made of the rotational velocity!
Response to Reviewer 3 comment 2: We have mentioned the effect of rotational velocities on microstructural evolution, mechanical performances, and corrosion resistance in the purpose of the study of the manuscript: “Microstructural evolution, mechanical performances, and corrosion resistance with different tool rotational velocities are investigated.” Furthermore, we have supplemented the description and study of rotational velocity in the introduction.
Comment 3: There are a few typos, as examples marked up in the PDF file.
Response to Reviewer 3 comment 3: Thank you for your suggestion, but unfortunately, we did not find the error markings you mentioned in the PDF.
Comment 4: No mention is made of the effect of rotational velocity on the mechanical and corrosion properties and this is an issue investigated in deep. The authors need to review the background of the rotational velocity to support the discussion of the results found.
Response to Reviewer 3 comment 4: We have made appropriate modifications to the introduction section. Zang et al. [26] successfully obtained 1060Al-SUS304 steel joints by FSW at a fixed rotational velocity of 800 rpm. Another research [27] has been conducted on the rotational velocities of 200 rpm, 600 rpm, and 1000 rpm for the 304SS-2219Al joint. Hua et al. [28] investigated the microstructure and properties of the 12Cr heat-resistant ferritic steel welded using FSW with different rotational velocities. They found that, at constant welding speed (200 mm/min), the optimal rotational velocities to produce sound welds should be from 400 rpm up to 800 rpm.
- Zhang, M.; Wang, Y.D.; Xue, P.; Zhang, H.; Ni, D.R.; Wang, K.S.; Ma, Z.Y. High-Quality Dissimilar Friction Stir Welding of Al to Steel with No Contacting between Tool and Steel Plate. Mater. Charact. 2022, 191, doi:10.1016/j.matchar.2022.112128.
- Zhang, M.; Liu, J.M.; Xue, P.; Liu, F.C.; Wu, L.H.; Ni, D.R.; Xiao, B.L.; Wang, K.S.; Ma, Z.Y. Eliminating Cu-Rich Intermetallic Compound Layer in Dissimilar Friction Stir Welding of 304 Stainless Steel and 2219 Al Alloy via Ultralow Rotation Speed. J. Mater. Process. Technol. 2024, 329, doi:10.1016/j.jmatprotec.2024.118444.
- Hua, P.; Moronov, S.; Nie, C.Z.; Sato, Y.S.; Kokawa, H.; Park, S.H.C.; Hirano, S. Microstructure and Properties in Friction Stir Weld of 12Cr Steel. Sci. Technol. Weld. Join. 2014, 19, 76–81, doi:10.1179/1362171813Y.0000000167.
Comment 5: Rotational velocity levels adopted must be presented, justified, and detailed in the materials and methods section, not only the range of velocities (line 78).
Response to Reviewer 3 comment 5: We have supplemented the materials and methods section with specific parameters used in this study: rotational velocities were 400 rpm, 500 rpm, 600 rpm, and 700 rpm.
Comment 6: Lines 82 and 83:There is no schematic drawing to show where the specimens were taken. It is necessary to prepare a scheme of the metallographic specimens' positioning. Besides, no information is given about the specimens of tensile tests (dimensions) and positioning.
Response to Reviewer 3 comment 6: We have included schematic diagrams of the positioning of metallographic specimens for the samples (Figure 2a), as well as schematic diagrams of the positioning of tensile test specimens (Figure 2a), and a schematic diagram depicting the dimensions of the tensile test specimens(Figure 2b), shown in Figure 2.
Comment 7: 3.3 No detail is provided about the specimen preparation for EBSD evaluation (Lines 91,and92).
Response to Reviewer 3 comment 7: In EBSD analysis, a Hitachi SU5000 SEM equipped with an Oxford probe was utilized. The operating voltage was set at 20 kV with a scanning step of 0.3 μm. The specimen preparation method for EBSD was identical to that used for metallographic specimens. A segment approximately 25 mm in length and 5 mm in width was cut perpendicular to the welding direction using electric wire cutting. This specimen was embedded in acrylic resin to create a metallographic specimen. The was polished to remove wire-cutting marks using 400-grit sandpaper, followed by stepwise grinding with 800, 1500, 2000, and 2500-grit sandpaper. Finally, the specimen was polished with 1.5 μm diamond polishing paste until the surface was scratch-free. The specimen was etched for 5 minutes using a corrosion solution with an HF: HNO3 with a volume ratio of 1:5:44. Electrolytic polishing was performed in a solution of 10 vol.% HClO4 + 90 vol.% CH3COOH at 25 V for 40 s. EBSD data were analyzed using HKL Channel 5 software. We have incorporated the specific details into the manuscript.
Comment 8: There is no detail about the electrochemical evaluation. No standard is presented, no electrochemical cell and configuration, and no scheme showing where the specimens were sliced out. It is mandatory to insert and describe the methods used.
Response to Reviewer 3 comment 8: We have incorporated the relevant details of the electrochemical tests into the manuscript. Electric wire cutting was used to cut samples from the center of the welded joints. Non-test areas were sealed with polymethyl methacrylate. The exposed area in the solution was 0.25 cm². A 3.5% NaCl aqueous solution was prepared as the testing solution, with a saturated calomel electrode as the reference electrode and a platinum sheet as the auxiliary electrode. The specimen served as the working electrode. Polarization curves were measured using a CHI 760E potentiostat. Electrochemical impedance spectroscopy was conducted using a PARSTAT 4000A single-channel potentiostat.
Comment 9: Hardness testing is proposed in section 2, but no result of hardness is presented in the manuscript.
Response to Reviewer 3 comment 9: We are sorry for that, this was a mistake in our manuscript writing process. We have removed the hardness testing from the materials and methods section.
Comment 10: From lines 109 to 126, the authors present the welding tool and too| surface images and argue that the tool was not significantly worn, indicating that the material adopted is good for FSW purposes. I don't think this is possible to say and the purpose of the work is not that. If the authors want to discuss this point, they must bring the background about it and adjust the work purpose.
Response to Reviewer 3 comment 10: This paragraph aims to demonstrate that the WC-W composites meet the basic requirements for welding high-melting-point metals such as stainless steel while being low-cost. The focus is not on claiming optimal wear resistance in welding. The manuscript has been appropriately revised in this section to ensure accuracy in its presentation
Comment 11: 4.2 Lines 127-128: “Figure 4 depicts the microstructures of different regions of the SZ. SZ was primarily composed of austenite and lath martensite, without visible ferrite phase and a phase". I think the images are in low magnification, hindering a confidence microstructure description. Also, considering the magnification, it is too risky to state: "without visible ferrite phase and a phase".
Response to Reviewer 3 comment 11: We have enlarged Figure 5 for a clearer and more intuitive observation. Additionally, we fully agree with your point that “stating without visible ferrite phase and a phase is too risky”. Therefore, we have removed this conclusion from the manuscript to ensure greater rigor.
Comment 12:From Figure 6 and Figure 7(EBSD results), becomes clear that a better description of the base metal (BM) should be done at the beginning of the manuscript. The readers need background about SUS301L(Austenitic steel), especially for the production history. It seems that the stee| strip is cold rolled, with 33% martensite in the microstructure (Figure 7). Is it really expected to have such a high amount of martensite in austenitic steels? How about the strain hardening and dislocation substructures? It needs further discussion and detailment. Also, the authors must describe the initial microstructure (BM),bring research works and references for that, and understand how important are these details to explain what is happening with the grain size and martensite fraction after welding, as different rotational velocities are chosen. Note, that the normal rule is to have grain size refinement from FSW welding in the stir zone, exactly the opposite of this work. This is the reason why this is very important to the clarity of the work.
Response to Reviewer 3 comment 12: (1) SUS301L stainless steel, as a metastable austenitic stainless steel, undergoes martensitic phase transformation induced by plastic deformation during cold rolling, which increases its strength and refines the grain size. Typically, grain growth and phase structure transformation occur after secondary processing. (2) The BM was obtained in the cold rolled condition. Due to cold rolling, BM has refined grains and contains a large amount of deformation-induced martensite. (3) Figures 9 show the kernel average misorientation (KAM) maps of the BM and the joint with different rotational velocities. The values of KAM were positively related to the density of dislocations. The average KAM (KAMavg) values of the joints increased with the increase of rotational velocities, while all the KAMavg values were lower than that of BM. This could be attributed to the reduction of martensite content and the growth of grain sizes. Dislocations were depleted during the transformation of martensite into austenite at the post-welding cooling stage. By contrast, the dynamic recovery related to the increased grain sizes in SZ also consumed dislocations. The KAMavg value of the dislocation was decreased under the combined influence of these two factors.
(4)Overall, above 400 rpm, the phase ratio changed and heat input rose, resulting in both grain size and martensite content increase with higher rotational velocities. (5)Typically, during the FSW process, severe plastic deformation and subsequent dynamic recrystallization induced by mechanical stirring result in finer grain sizes. However, if the BM possesses ultrafine grain sizes or if excessive heat input occurs during welding, the grain size in the SZ may be larger than the original grain size of the base material.
Comment 13: Grain Size(Figure 6f): Could the authors insert the number of analyzed grains in each evaluated area? How many images were analyzed to have the average value? It seems that the grain size is similar for all welding conditions, roughly about 9 micrometers. I think discussing the effect of welding rotational velocity on the grain size is a bit hard in the present case. The authors could try to differentiate the welding conditions based on the grain size heterogeneities.
Response to Reviewer 3 comment 13:(1) The number of analyzed grains in each evaluated area is as follows: BM:1273, 400 rpm:3411, 500 rpm:4021 600 rpm:6051, and 700 rpm:8561. (2) After analyzing three images for each parameter, we calculated the average. (3) We observed minimal differences in grain size under the four experimental parameters as shown in Figure 7f. However, it qualitatively reflects the trend of grain size variation with continuous rotational velocity.
Comment 14: How many tensile specimens were tested? Here, once again, is important to bring the details pointed out in item 3.2 of this report.
Response to Reviewer 3 comment 14: Each parameter was tested on three tensile specimens, and the average values were taken. We have added detailed information about this in the manuscript.
Comment 14: I could not find the discussion section. The results still need better analysis and the discussion needs to bring the background, citing previous works and debating what is happening in the specific case here with the microstructure, mechanical properties, and corrosion behavior. This is missing, to a certain point, because the introduction did not focus on the effect of rotational velocities on the microstructure, properties, and corrosion behavior. It should be re-written and previous works must be there. Thus, to discuss the results will be a bit easier.
Response to Reviewer 3 comment 14:. We have supplemented the discussion in the manuscript regarding rotational velocity, providing examples from Zang et al.[26, 27]and Hua et al.[28], covering arguments on its influence on microstructure, mechanical properties, and corrosion resistance. We have rewritten the conclusion section of the paper.
- Zhang, M.; Wang, Y.D.; Xue, P.; Zhang, H.; Ni, D.R.; Wang, K.S.; Ma, Z.Y. High-Quality Dissimilar Friction Stir Welding of Al to Steel with No Contacting between Tool and Steel Plate. Mater. Charact. 2022, 191, doi:10.1016/j.matchar.2022.112128.
- Zhang, M.; Liu, J.M.; Xue, P.; Liu, F.C.; Wu, L.H.; Ni, D.R.; Xiao, B.L.; Wang, K.S.; Ma, Z.Y. Eliminating Cu-Rich Intermetallic Compound Layer in Dissimilar Friction Stir Welding of 304 Stainless Steel and 2219 Al Alloy via Ultralow Rotation Speed. J. Mater. Process. Technol. 2024, 329, doi:10.1016/j.jmatprotec.2024.118444.
- Hua, P.; Moronov, S.; Nie, C.Z.; Sato, Y.S.; Kokawa, H.; Park, S.H.C.; Hirano, S. Microstructure and Properties in Friction Stir Weld of 12Cr Steel. Sci. Technol. Weld. Join. 2014, 19, 76–81, doi:10.1179/1362171813Y.0000000167.
Comment 15: This section is not responding to the work purpose, since:
The microstructure needs to be analyzed in depth. From the images (Figure 4) is not possible to conclude about austenite-to-martensite and deleterious phase presence. Also, the authors need to bring the previous description of BM and all production details.
The grain size should be discussed more because the trend of grain refinement in the stir zone of FSW was not observed in the present work. Why?There is a missing corrosion test description, as well as, the possible reasons for the behavior, also dependent on a better microstructure description.
Response to Reviewer 3 comment 15: (1) We adjusted the magnification of Figure 4 (as shown in Figure 5) to allow for a more intuitive observation of the microstructural distribution.
(2) We have added the descriptions of the BM in the Introduction section. In the materials and methods, we have added specific rotational velocities parameters and the parameters of all tests. (3) The BM possesses ultrafine grain sizes or if excessive heat input occurs during welding, the grain size in the stirred zone (SZ) may be larger than the original grain size of the BM. (4) We have provided the details of the corrosion tests in the second section and included a discussion on rotational velocity and corrosion resistance in the results section.
